# Transient dynamics of the phase transition in VO$_2$ revealed by mega-electron-volt ultrafast electron diffraction

Chenhang Xu [1], Cheng Jin[2,3], Zijing Chen[2,3], Qi Lu[1], Yun Cheng[2,3], Bo Zhang[1], Fengfeng Qi[2,3], Jiajun Chen[1], Xunqing Yin[1], Guohua Wang[1], Dao Xiang [2,3,4,5] ✉ & Dong Qian [1,4,6] ✉

Vanadium dioxide (VO$_2$) exhibits an insulator-to-metal transition accompanied by a structural transition near room temperature. This transition can be triggered by an ultrafast laser pulse. Exotic transient states, such as a metallic state without structural transition, were also proposed. These unique characteristics let VO$_2$ have great potential in thermal switchable devices and photonic applications. Although great efforts have been made, the atomic pathway during the photoinduced phase transition is still not clear. Here, we synthesize freestanding quasi-single-crystal VO$_2$ films and examine their photoinduced structural phase transition with mega-electron-volt ultrafast electron diffraction. Leveraging the high signal-to-noise ratio and high temporal resolution, we observe that the disappearance of vanadium dimers and zigzag chains does not coincide with the transformation of crystal symmetry. After photoexcitation, the initial structure is strongly modified within 200 femtoseconds, resulting in a transient monoclinic structure without vanadium dimers and zigzag chains. Then, it continues to evolve to the final tetragonal structure in approximately 5 picoseconds. In addition, only one laser fluence threshold instead of two thresholds suggested in polycrystalline samples is observed in our quasi-single-crystal samples. Our findings provide essential information for a comprehensive understanding of the photoinduced ultrafast phase transition in VO$_2$.

Ultrafast photoexcitation by femtosecond (fs) lasers can help understand and manipulate the interaction among electronic, magnetic, and structural degrees of freedom[1–5]. Various quantum materials, including charge-density-wave materials[6–11], superconductors[12–15], ferromagnets[16], and phase transition materials[17–21], have been explored via ultrafast photoexcitation

techniques. Of these, the strongly correlated oxide VO$_2$ has attracted extensive interests. VO$_2$ exhibits a first-order insulator-to-metal transition accompanied by a monoclinic-to-rutile structural phase transition (SPT) upon heating[22,23]. The transition occurs near the room temperature with a dramatic change in resistance as well as in optical properties. Such a transition can also be induced by

[1]Key Laboratory of Artificial Structures and Quantum Control (Ministry of Education), Shenyang National Laboratory for Materials Science, School of Physics and Astronomy, Shanghai Jiao Tong University, Shanghai 200240, China. [2]Key Laboratory for Laser Plasmas (Ministry of Education), School of Physics and Astronomy, Shanghai Jiao Tong University, Shanghai 200240, China. [3]Collaborative Innovation Center of IFSA, Shanghai Jiao Tong University, Shanghai 200240, China. [4]Tsung-Dao Lee Institute, Shanghai Jiao Tong University, Shanghai 200240, China. [5]Zhangjiang Institute for Advanced Study, Shanghai Jiao Tong University, Shanghai 200240, China. [6]Collaborative Innovation Center of Advanced Microstructures, Nanjing 210093, China. ✉e-mail: dxiang@sjtu.edu.cn; dqian@sjtu.edu.cn

ultrafast photoexcitation[17–20,24–37]. Thus, VO$_2$ has great potential in fast/smart thermal switchable devices and photonic applications.

In the monoclinic insulating phase (called the M$_1$ phase), V atoms dimerize and form zigzag chains (called the superstructure in this work). In the rutile (tetragonal) metallic phase (called the R phase), V–V dimers and related zigzag chains disappear. The unit cell in the M$_1$ phase is nearly twice larger than that in the R phase, resulting in more X-ray or electron diffraction peaks (called superstructure peaks) in the M$_1$ phase. To understand the phase transition, VO$_2$ was studied by various experimental methods, such as terahertz (THz) spectroscopy[26–28,30,31], optical spectroscopy[24,25], X-ray absorption spectroscopy (XAS)[36], ultrafast electron diffraction (UED)[17–19,35,37,38], and ultrafast X-ray diffraction (UXRD)[20]. However, the true nature of the phase transition is still not yet fully understood. For instance, while the SPT involves both the loss of the V–V dimerization and the lattice transformation[39,40], the related atomic pathway has not been fully revealed[17,37].

The very first UED (in a reflected electron diffraction mode) experiment with femtosecond resolution[17] on single crystalline VO$_2$ proposed that V–V dimers first dilate to form an intermediate phase without dimers but with V zigzag chains. Then V–V bonds begin tilting to eliminate the zigzag chain, leading to the R-phase[17]. UED experiment[19] on polycrystalline VO$_2$ proposed that V atoms might first reach an intermediate structure, which is akin to the M$_2$ phase (a phase with a special configuration of V zigzag chains[41]). UXRD experiment have found that the disordered movement of V atoms played an important role in the light-induced ultrafast phase transition[20]. Recent UED experiment on single-crystal VO$_2$ indicated that V–V dimers dilate and tilt simultaneously in a coherent way in the very early stage of the photoinduced SPT (PSPT)[37].

SPT from the M$_1$ to R phase will not only change the fractional coordinates of V atoms within a unit cell, but also cause a change in the lattice symmetry[39]. VO$_2$ has the monoclinic symmetry in the M$_1$ phase, and turns into the tetragonal symmetry in the R phase. So far, the question whether the change of the fractional position of V atoms coincides with the change of the lattice symmetry in PSPT remains unanswered. An ultrafast diffraction conoscopy experiment showed that VO$_2$ films grown on an Al$_2$O$_3$ substrate could sustain an intermediate biaxial structure that is different from the initial M$_1$ phase (the M$_1$ phase is also a biaxial phase) for several hundreds of femtoseconds, which provides some indirect information that the disappearance of V–V dimers and zigzag chains may not coincide with the change of the crystal symmetry in PSPT[34].

Furthermore, previous UED experiments on polycrystalline VO$_2$ samples suggested a photoexcited long-lived monoclinic metallic phase (called the mM phase) lasting for hundreds of picoseconds[18]. This phase is isostructural to the M$_1$ phase[18,35,36,38]. The key evidence in this UED experiment was that the laser fluence threshold for non-superstructure peaks is smaller than that for superstructure peaks[18]. However, the experimental uncertainty is considerably large due to the low signal-to-noise ratio in polycrystalline samples. Recently, only one threshold was observed in time-resolved XAS[36]. UED on a single crystalline VO$_2$ sample thinned by focused ion beam from a bulk single-crystal also gave some indication of one threshold[37].

In this work, we succeed in fabricating high quality quasi-single crystalline VO$_2$ freestanding films and revisit the PSPT using mega-electron-volt (MeV) UED with 50 fs resolution[42]. Due to the high signal-to-noise ratio and the domain structures in quasi-single crystalline samples, we reveal the comprehensive atomic pathway, including the melting of V–V dimers and the transformation from monoclinic symmetry to tetragonal symmetry. We present direct evidence that the disappearance of superstructure does not coincide with the change of the crystal symmetry. We find that V–V dimers melt and V atoms reach the same fractional coordinates as that in the R phase within 200 fs, resulting in a transient monoclinic structure without V–V dimers and zigzag chains. Then, it continues to evolve from this transient monoclinic structure to the final tetragonal structure within 5 picoseconds. In addition, the laser fluence thresholds are accurately measured benefited from the well separation of the superstructure peaks from other diffraction peaks. A single threshold is obtained within the experimental uncertainty.

## Results

### Preparation of quasi-single crystalline freestanding VO$_2$ films

The state-of-the-art UED technique uses transmission mode, so free-standing films are required. Recently, Sr$_3$Al$_2$O$_6$ (SAO) has been used as a water-soluble sacrificial layer to obtain freestanding single crystalline films prepared by pulsed laser deposition (PLD)[43]. Directly growing ultrathin freestanding single-crystal films on SAO has been demonstrated for materials with perovskite structures such as SrTiO$_3$ (STO)[43], La$_{0.7}$Sr$_{0.3}$MnO$_3$[43], BaTiO$_3$[44], BiFeO$_3$[45], and LaMnO$_3$[46] that have lattice parameters close to SAO. However, this method does not apply to VO$_2$ due to the large lattice mismatch. Our solution for obtaining free-standing VO$_2$ films by PLD is to add an ultrathin STO(111) buffer layer between the VO$_2$ and SAO. By limiting the thickness of STO buffer layer to 2 nm, we can minimize the effect from the buffer layer on our diffraction measurements. We grew 20 nm SAO on a bulk STO(111) substrate followed by a 2 nm single-crystal STO(111) film as the buffer layer for the VO$_2$ growth. Then we grew ~40 nm VO$_2$ on the STO(111) buffer layer. The layout of the sample is shown in Fig. 1a.

The epitaxial growth of VO$_2$ films on the STO(111) surface was systematically studied in the previous report[47]. The normal direction of the film is [010], which nicely matches the UED experiments because V–V dimers exist in the (010) plane. By the strict definition, VO$_2$/STO(111) film is a (010) textured film. Unlike usual textured films with lots of in-plane orientations, VO$_2$/STO(111) film only has three in-plane orientations (called three 120° domains, for convenience) in the R phase. In the M$_1$ phase, each domain will further form twins. Details about the crystal structure are presented in Supplementary Note 1 in the Supplementary Materials (SM). To be distinct from usual textured films, we call VO$_2$/STO(111) films quasi-single crystalline films. After removing SAO by etching in water, we obtained relatively large flakes of freestanding 40-nm VO$_2$/2-nm STO films. The lateral size of the flakes varies around 0.5 × 0.5 mm. See more details about the samples in Supplementary Fig. 1. The freestanding films are transferred to a transmission electron microscope (TEM) grid with ultrathin amorphous carbon film for MeV UED measurements. The amorphous carbon film effectively reduces heat accumulation from the optical pump with a repetition rate of 100 Hz.

### Static electron diffractions of quasi-single-crystal VO$_2$

The static electron diffraction pattern at 300 K (M$_1$ phase) measured by 200 keV TEM is shown in Fig. 1c with Miller indices shown for several representative diffraction spots. All Miller indices use the M$_1$-phase's representation. Diffraction spots from the 2-nm STO buffer layer were also observed, and are indicated by "STO". Figure 1d shows the simulated ideal TEM diffraction pattern in the M$_1$ phase considering three 120° domains and the twinning in each domain. The twinning in the M$_1$ phase results from the difference in the lattice constants (a$_M$ = 5.356 Å and c$_M$ = 5.383 Å[39,48]) (see more details in Supplementary Figs. 1 and 2). Red/green/black colors in Fig. 1d represent the contributions from three 120° domains, respectively. The simulation is in good agreement with the measurements. The elongated shape along the tangential direction in Fig. 1c is related to two effects. First, three sets of diffraction spots from 120° in-plane domains distribute in the tangential direction, seen from Fig. 1d. They partially overlapped in experiments, which makes the spots broaden along the tangential direction. Second, the shape of crystal grains is anisotropic (see details in Supplementary Fig. 1), which also makes the spots broader in the tangential direction than in the radial direction.

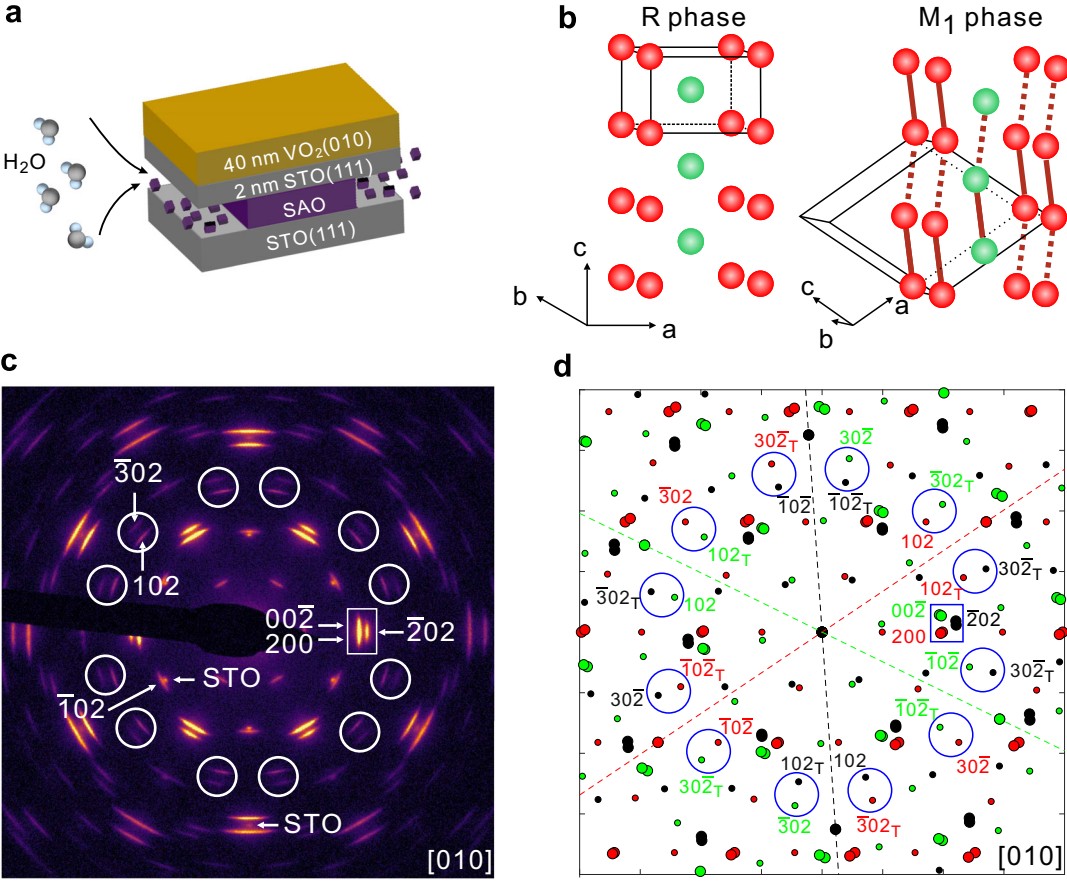

**Fig. 1 | Static electron diffractions of VO₂ by TEM. a** Schematic layout of the VO₂ films for UED experiments. STO represents SrTiO₃(111) which is used as substrate and buffer layer. SAO represents Sr₃Al₂O₆ which is used as sacrificial layer. SAO can be etched by water indicated by black arrows. **b** Crystal structures of VO₂ in the R phase and M₁ phase. Only V atoms are presented. Red solid lines in the M₁ phase represent V–V dimers. Hexahedrons with black lines are the unit cells. **c** Static electron diffraction pattern measured at 300 K by TEM. The incident electron beam is along [010] direction of VO₂. **d** Simulated TEM diffraction pattern in the M₁ phase with three domains and twinning. Red/green/black colors represent the contributions from three 120° domains, respectively. For convenience, we use smaller dots to present the superstructure spots. The subscript "T" means the contribution from a twin domain.

In Fig. 1c, we marked 12 groups of spots that are well separated from STO spots by white circles. There are two spots in each group. Indexed in Fig. 1d, the inner 12 spots are $102/102_T/\bar{1}0\bar{2}/\bar{1}0\bar{2}_T$. Each 120° domain contributes four spots when twinning is considered. The subscript "T" means the contribution from a twin domain. 102 and $102_T$ are equivalent because of the same index despite that they are from different domains. Since VO₂ has the inversion symmetry, 102 and $\bar{1}0\bar{2}$ are equivalent. So, 102, $102_T$, $\bar{1}0\bar{2}$ and $\bar{1}0\bar{2}_T$ are indeed equivalent spots. The outer 12 spots are $30\bar{2}/30\bar{2}_T/\bar{3}02/\bar{3}02_T$. They are also equivalent spots. Therefore, such 12 groups are equivalent. Another important group of spots is marked by a white rectangle in Fig. 1c. According to Fig. 1d, this group consists of six spots, i.e., $00\bar{2}, \bar{2}00_T, 200, 00\bar{2}_T, \bar{2}02$ and $20\bar{2}_T$. Due to the limited space, we omitted indices of $\bar{2}00_T, 00\bar{2}_T$ and $20\bar{2}_T$ for simplicity. Each 120° domain contributes two spots when twinning is considered.

**Static electron diffractions of quasi-single-crystal VO₂ by UED**

Figure 2a presents the static diffraction pattern measured by the 3-MeV electrons in the MeV UED instrument below and above the phase transition temperature. It should be noted that three 120° domains equally contribute to the diffraction intensities because the electron beam size (~150 μm in diameter) in UED is much bigger than the grain size in the film. The momentum resolution in Fig. 2a is not as good as that in Fig. 1c. There are two reasons. First, unlike the TEM, the electron beam is not focused in UED to achieve a high temporal resolution. Second, the higher kinetic energy in the MeV UED also

slightly lowers the momentum resolution. As a result, the diffraction spots close to each other in Fig. 1c merge together in Fig. 2a. For instance, 102 and $\bar{3}02$ become one spot. For simplicity, we use 102 to index this big spot. 102 and $\bar{3}02$ spots are only existed in the M₁ phase related to the V–V dimers. They are superstructure spots. The structure factor of 102 and $\bar{3}02$ are zero in the R phase, so they both disappear during the SPT from the M₁ phase to R phase. Similarly, $00\bar{2}$, 200, and $\bar{2}02$ spots are also overlapped. We use 200 as the simple index. In the previous diffraction experiments in a single crystalline sample[37], the intensities of $00\bar{2}$, 200, and $\bar{2}02$ spots all enhance during the SPT from the M₁ phase to R phase. Therefore, it is reasonable to trace the intensities of the big 102 and 200 spots to study the dynamics of the SPT.

As expected, the superstructure spots disappear above the transition temperature (Fig. 2a right). Figure 2b presents the line profiles of a superstructure peak along the direction indicated by the white line in Fig. 2a. At 300 K, the diffraction peak is well-defined. At 345 K, the diffraction peak disappears and the background is slightly enhanced due to the thermal effect. To simulate the polycrystalline sample to some extent, we plot the azimuthally averaged diffraction intensity in Fig. 2c. The superstructure peaks (100, $\bar{1}02$ and 102) are influenced by the neighboring non-superstructure peak (200) with very high intensity, which renders accurate determination of the threshold difficult in polycrystalline samples[18]. Following the temperature dependence of the intensity of 102 peak, a clear SPT with a temperature hysteresis loop was observed (see details in Supplementary Fig. 3).

 

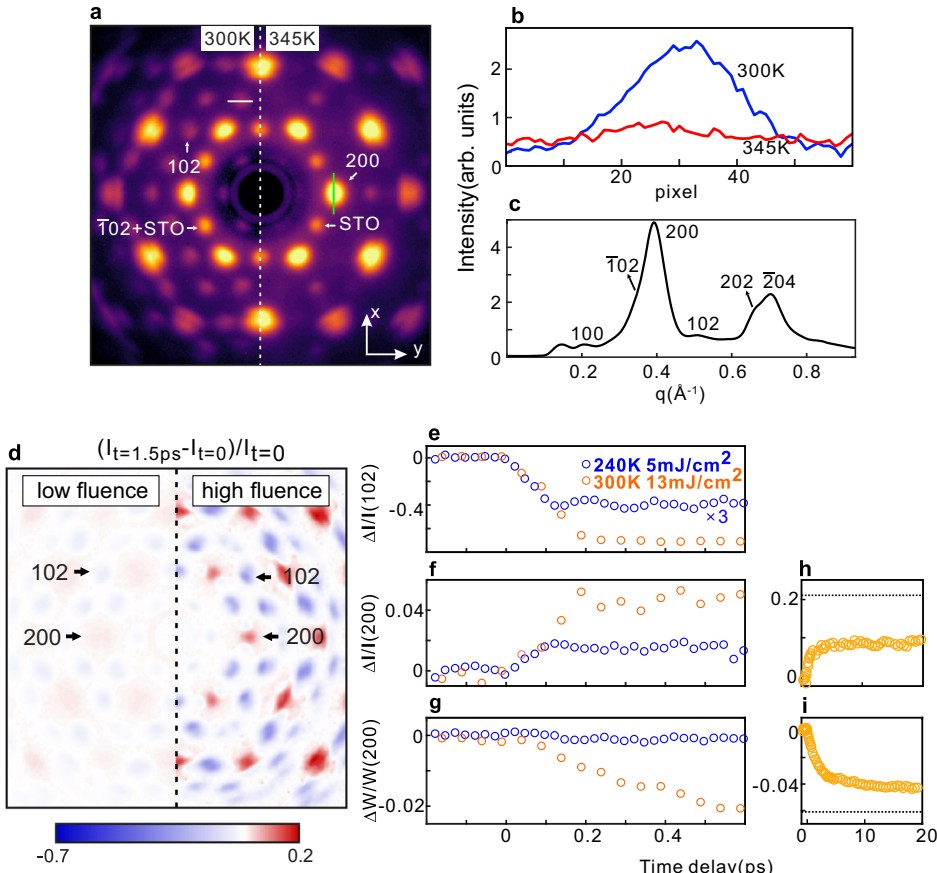

**Fig. 2 | Ultrafast evolution of the diffraction spots in VO₂. a** Static electron diffraction patterns measured at 300 K (left) and 345 K (right) with the 3-MeV electrons in UED. Green line labels the direction where we trace the peak width. **b** Line profile along the white line in **a** at 300 K and 345 K, respectively. **c** Azimuthally averaged intensity of the diffraction pattern at 300 K. **d** Image plot of the intensity difference between the diffraction patterns at $t = 1.5$ ps after photoexcitation and $t = 0$ (before photoexcitation) under a low fluence (laser fluence is 5 mJ/cm² at 240 K) and a high fluence (laser fluence is 13 mJ/cm² at 300 K). Time evolution of the intensity for (**e**) 102 peaks, (**f**) 200 peaks and (**g**) width of 200 peaks. Data up to 20 ps for (**h**) intensity of 200 peaks and (**i**) width of 200 peaks under a high fluence. Black dashed line marks the level of relative change when 100% $M_I$-R transition occurs based on the thermal transition data.

## Ultrafast evolution of the diffraction spots in VO₂

The VO₂ sample shows an ultrafast response to the laser pump. Figure 2d presents the normalized intensity differences between the diffraction pattern at $t = 1.5$ ps after photoexcitation and that at $t = 0$ (without photoexcitaion) under the low (-5 mJ/cm² at 240 K) and high (-13 mJ/cm² at 300 K) laser fluence, respectively. Upon photoexcitation, the intensity of all the superstructure spots decreases, while non-superstructure spots enhance.

Figure 2e shows the transient intensity change of the 102 peak. Under a low fluence, the intensity only drops by about 13% and oscillations of 6 THz related to a coherent phonon of the zigzag motion of V atoms[20,25,29,49], are clearly observed. There is no SPT under low fluence (see data with 1.5 ps in Supplementary Fig. 4). A similar coherent phonon was detected in the UXRD experiment[20], but had not been observed in previous UED experiments primarily due to the insufficient temporal resolution. In contrast, under a high fluence, the intensity drops by more than 70% within 200 fs, and no oscillation was detected. The absence of oscillation implies a fast structural transition, in agreement with the UXRD experiments[20]. Qualitatively consistent with previous UED experiments[18], the intensity of the non-superstructure 200 peak increases after photoexcitation, as shown in Fig. 2f. However, the timescale dichotomy in the intensity evolution of the superstructure and non-superstructure diffraction peaks observed in polycrystalline samples[18,35,38] is not observed in our measurements. Rather, the intensities of all the peaks changed with nearly the same

femtosecond timescale, also consistent with UXRD on single-crystal samples[20].

Interestingly, we can trace the ultrafast structure evolution by following not only the peak intensity but also the peak width due to the high signal-to-noise ratio and the domain structure in our quasi-single crystalline samples. Presented in Fig. 2g, the peak width of 200 ($W_{200}$) along the x-direction (indicated by a green line in Fig. 2a) decreased with a picosecond timescale under a high fluence, while it remained constant under a low fluence. To do the analysis on the peak width along x-direction, we superposed the intensities of 200 spot along the y-direction. Since no SPT occurs under a low fluence, the decrease of $W_{200}$ under a high fluence is very likely related to the SPT. In fact, the decrease of $W_{200}$ can be understood based on the difference in the lattice constants $a_M$ and $c_M$, as well as the angle (β) between $a_M$ and $c_M$, in the monoclinic structure. As we show in Fig. 1, there are three 120° domains plus twin domains in our samples, which has two implications for the peak width. First, the separation between the diffraction spots from different 120° domains is correlated to β. Second, the twin domains originated from the difference in $a_M$ and $c_M$ cause the additional peak splitting. With the gradual suppression of twin domains and increase in β, six diffraction spots (Fig. 1d) inside a big 200 spot get closer and closer (see more details in Supplementary Fig. 5). As a result, the peak width becomes narrower. Figure 2i indicates that the peak width decreases in a picosecond timescale, implying that a monoclinic structure persists for several picoseconds even after the complete disappearance of the superstructure in VO₂.

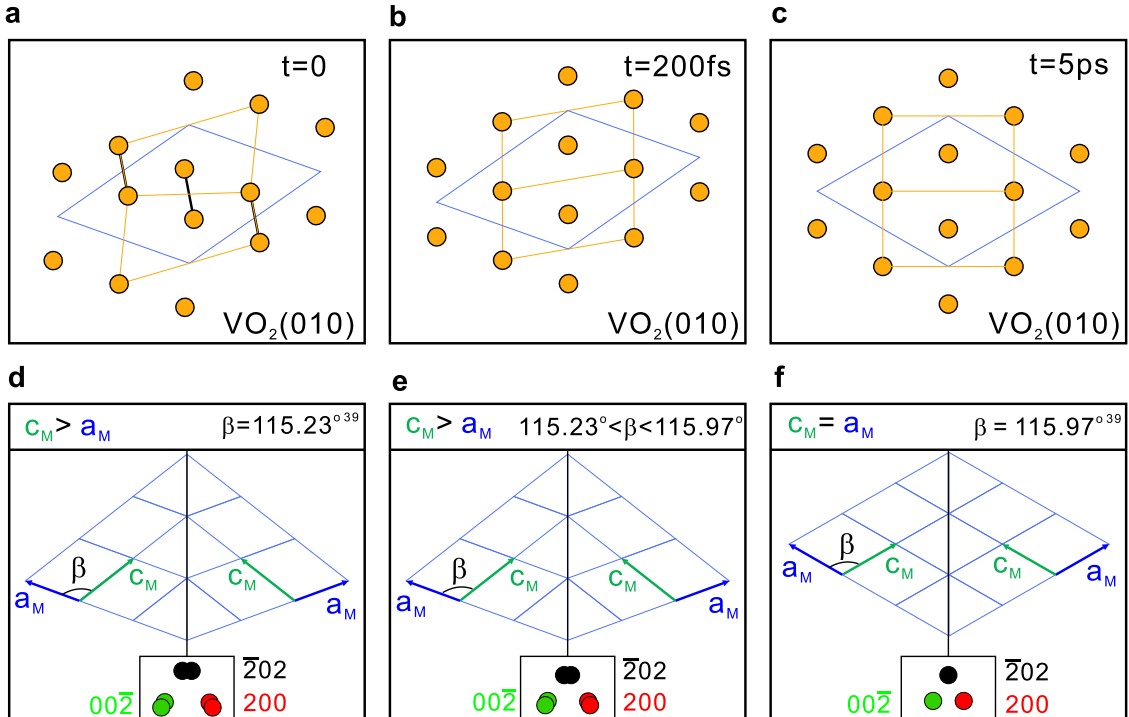

**Fig. 3 | Transient monoclinic structure without V−V dimers in VO₂.** Schematic of Monoclinic $M_1$ phase with V−V dimers and zigzag chains at $t = 0$ (**a**), transient monoclinic structure without V−V dimers and zigzag chains at $t = 200$ fs (**b**) and R phase at $t = 5$ ps (**c**). Orange dots represent the projected positions of V atoms on the (010) plane. Black lines present the V−V dimers. Orange lines are a guide for the eyes. Blue quadrilateral indicates the unit cell using $M_1$-phase's representation. The sketches of twin domains at $t = 0$ (**d**), $t = 200$ fs (**e**), $t = 5$ ps (**f**) after photoexcitation. Red\green\black dots indicate the diffraction spots from three 120° domains and the twin domains respectively. And the separation between the two spots is exaggerated for clarity.

## Transient monoclinic structure without V−V dimers

Based on our experimental results, we propose that there are two processes with different timescales during the transition from the $M_1$ Phase to R phase in the PSPT of VO₂. Figure 3a–c sketches the projected positions of V atoms on the (010) plane at three different time. The corresponding sketches of twin domains are shown in Fig. 3d–f. The insets present six diffraction spots inside the big 200 spot observed in Fig. 2a. In the initial $M_1$ phase (Fig. 3a, d), the sample consists of the superstructure and twin domains. In the first process, the V−V dimers dilate and tilt simultaneously, which is characterized by the very fast disappearance of the superstructure peaks (Fig. 2e). This fast process has been observed in previous scattering studies and was thought to end up in the R phase in a femtosecond timescale[18,20]. It should be noted that in diffraction experiments, the intensity of diffraction peaks is mainly affected by the fractional coordinates of atoms in the unit cell. Therefore, the disappearance of the superstructure peaks does not necessarily mean that VO₂ is in the R phase, but rather it means that V atoms are in the same fractional coordinates as in the R phase.

The evolution of the width of 200 peak in Fig. 2g indicates that the fast process does not lead to the R phase, but to an intermediate monoclinic structure without V−V dimers and/or zigzag chains. Figure 3b, e sketches the final stage of this fast process at $t = 200$ fs. The V−V dimers and/or zigzag chains no longer exist, and V atoms are located at the same fractional coordinates as in the R phase. Correspondingly, there is no superstructure spot, but $a_M$ and $c_M$ remains different and β does not reach that in the R phase. Hence, the twin domains still exist after the first process. The second process is the transition from the monoclinic symmetry to the tetragonal symmetry through the lattice expansion, mainly along the $a_M$ direction. This process is characterized by the continual decrease of $W_{200}$ (Fig. 2g). It is a slow process and nearly finishes within 5 ps, ending up in the R

phase sketched in Fig. 3c, f with $a_M = c_M$ and no twinning. The second process is not distinguishable from the first process within the first 200 fs because the movement of V atoms in the first process expands $a_M$ simultaneously. The second process suggests that a transient monoclinic structure without V−V dimers and/or zigzag chains exists after the first process and lasts for several picoseconds.

## Fluence thresholds measurements

To determine the fluence thresholds for the PSPT, we measured the diffraction intensity at a time delay of 10 ps when the electron and lattice largely reaches a quasi-equilibrium condition. Figure 4a shows the fluence dependence of 102 peak measured at 300 K. A distinct kink is observed at ~3.1 mJ/cm². The intensity decreases slowly below this value, while a fast decrease is observed above this value. Because the 102 peak is directly associated with the V−V dimerization, the presence of a kink indicates a fluence threshold for SPT, e.g., a critical energy is needed to melt the V−V dimers for the transition to the R phase. Therefore, at 300 K, we have the threshold ($F_{c1}$) of ~3.1 mJ/cm². For the 200 peak, a kink is also observed at ~3.1 mJ/cm², as shown in Fig. 4b. Thus, we obtained the threshold ($F_{c2}$) of ~3.1 mJ/cm². This is in stark contrast to previous work in polycrystalline samples where $F_{c1}$ is much larger than $F_{c2}$[18]. We repeated the measurements at several temperatures (280, 260, 240, and 220 K) and the threshold for the 102 and 200 peaks are consistently very similar (see details in Supplementary Fig. 6). We also measured the fluence threshold for the change of $W_{200}$ as shown in Fig. 4c. Again, it has a similar threshold ($F_{c3}$) of ~3.1 mJ/cm².

It should be noted that the diffraction measurements do not yield the direct information of free-electron density that confirm metallicity. Ref. [18] suggested an interesting photoinduced mM phase with a very long lifetime (sever hundreds of picoseconds). Besides the optical measurements, one of the key evidences is the existence of two thresholds in UED. Since only one threshold is detected in our

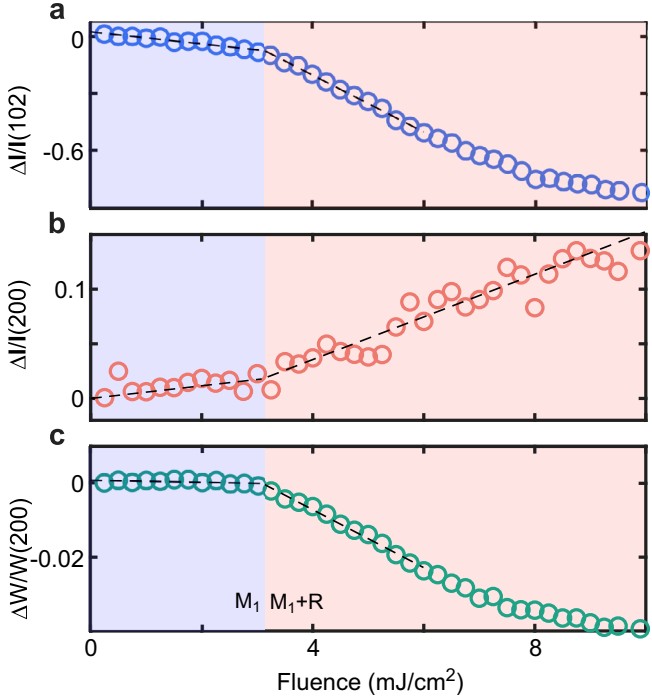

**Fig. 4 | Measurements of the laser fluence.** Fluence dependence of the changes measured at $t = 10$ ps after photoexcitation for 102 peak intensity (**a**), 200 peak intensity (**b**), and 200 peak width (**c**). Dashed lines are a guide to the eye.

experiments, the proposed long lifetime mM phase in ref. [18] very likely does not exist in our quasi-single crystalline samples.

## Discussion

The pathway with two processes we proposed during the PSPT in $VO_2$ differs from the proposed two steps model[17]. In the previous model, the first step is the dilation of the V–V dimers, followed by the tilting of the V–V bonds. Therefore, two timescales were expected for the decay of the superstructure peaks. However, only one timescale following the superstructure peak was observed by the UXRD[20] and UED[37] experiments in the single crystalline samples. While two timescales were observed by UED in the polycrystalline $VO_2$ samples, they were extracted from superstructure and non-superstructure peaks[18]. Some optical measurements observed two processes with two timescales of <10 ps and >45 ps[50] which is not directly related to the ultrafast phase transition. In our experiments, only one timescale was obtained based on the superstructure peak, which is in agreement with the UXRD and the UED experiments in the single-crystal samples. A single timescale in the superstructure peak indicates a direct movement of V atoms; in other word, the V–V dimers dilate and tilt at the same time to the fractional coordinates the same as that of the R phase, which is the first process in our proposal. The second process revealed in our experiments is the transformation from a transient monoclinic structure to the tetragonal R phase. Recent ultrafast X-Ray hyperspectral imaging experiments[40] observed two time constants of 132 fs and 4.75 ps, and they are not related to dilation and tilting[17]. Considering the fact that these time constants are close to the timescales we observed, it is possible that they might be related to the two processes revealed in our experiments, which would be interesting to explore in the future.

The second process can hardly be explored in polycrystalline samples by UED. Although we found this new process in quasi-single crystalline samples, we still cannot tell the exact route how V atoms move as a function of time because of the in-plane domains. High quality freestanding single crystals are needed to completely solve this

problem. Preliminary results have been obtained in single-crystal samples[37]. However, much more efforts are needed to have well controlled freestanding single crystalline samples. In principle, by tracing the intensity of various diffraction spots in single crystals, we can determine the position of V atoms through the structure factor.

In summary, we studied the PSPT in the quasi-single crystalline $VO_2$ samples with the MeV UED. The highly improved signal-to-noise ratio and temporal resolution not only allowed us to observe the coherent phonon related to the zigzag motion of V atoms connecting the $M_1$ and R phases under a low laser fluence, but also revealed two processes with different timescales when SPT occurs: the fast destruction of V–V dimers and zigzag chains within ~200 fs and the relatively slow transformation from the monoclinic symmetry to the tetragonal symmetry within ~5 ps. Only a single fluence threshold was observed for different diffraction peaks. Our work promotes the comprehensive understanding of the ultrafast PSPT in $VO_2$ and will stimulate more theoretical and experimental efforts in the future.

## Methods

### Sample preparation

Quasi-single-crystal $VO_2$ films were epitaxially grown on $SrTiO_3$ (111) (STO) substrates by pulsed laser deposition with a KrF excimer laser ($\lambda = 248$ nm, Coherent) using a $V_2O_5$ target. The base pressure of our PLD system is $8 \times 10^{-10}$ Torr. The STO(111) substrate was first annealed at 1000 °C for 1 h in oxygen at $1 \times 10^{-6}$ Torr to obtain a clean and flat surface. Then, a 20-nm SAO (111) sacrificial layer followed by a 2-nm STO (111) buffer layer was deposited at 880 °C under $5 \times 10^{-6}$ Torr oxygen pressure. The thickness of SAO and STO was monitored by the reflection high energy electron diffraction (RHEED) oscillations. The laser repetition rate is 5 Hz, and the fluence is 1.5 J/cm². After that, 40 nm quasi-single-crystal $VO_2$(010) films was deposited on the STO (111) buffer layer at 450 °C in 3 mTorr oxygen pressure. The laser repetition rate is 10 Hz, and the fluence was 1 J/cm². We used deionized water to remove the SAO layer and transferred the 40 nm $VO_2$(010)/ 2 nm STO (111) freestanding film to a TEM grid with ultrathin (~5 nm) amorphous carbon film for 200 keV TEM and 3 MeV UED measurements. Thickness of $VO_2$ film were checked by AFM.

### Transmission electron microscopy

The TEM experiments were conducted using a Talos F200X STEM. The accelerating voltage was 200 kV. The selected area aperture is 200 μm in diameter and the diffracting region is 4 μm in diameter. The electron beam was incident on the sample along the [010] direction of $VO_2$.

### MeV ultrafast electron diffraction

The electron beam in this MeV UED system has a kinetic energy of about 3 MeV and is compressed with a double-bend achromatic magnet system. An electron-multiplying CCD camera and a phosphor screen were used to measure the diffraction pattern. A 800-nm Ti:Sapphire laser with a pulse width of ~30 fs was used as the pump laser. The radius spot size of the pump laser was 1.5 mm. The pump laser was near-normal incidence. More details of the MeV UED setup can be found in the literature[42]. A repetition rate of 100 Hz or 50 Hz was used to mitigate the heat accumulation. The temporal resolution was about 50 fs (FWHM) for the measurements of the ultrafast dynamic. The temporal resolution was 100 fs to increase the signal-to-noise ratio for the measurements of the thresholds.

## Data availability

All of the data supporting the conclusions are available within the article and the Supplementary Information. Because the raw data include tens of thousands of images, they are available from the corresponding authors upon request.

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

## Acknowledgements

This work was supported by the National Key R&D Program of China (No. 2022YFA1402400, No. 2021YFA1400202, No. 2021YFA1400100), the National Natural Science Foundation of China (Grants No. 11521404, No. 11925505), the office of Science and Technology, Shanghai Municipal Government (No. 16DZ2260200), and the additional support from a Shanghai talent program.

## Author contributions

D.Q. and D.X. conceived the experiments, C.X. conducted the experiments with the help of C.J., Z.C., Q.L., Y.C. B.Z., F.F.Q., J.C., X.Y., G.H.W., C.X., D.Q. and D.X. analyzed the results. All authors reviewed the manuscript.

## Competing interests

The authors declare no competing interests.
