## [Peer Review File · Nature Communications]

Transient dynamics of the phase transition in VO₂ revealed by mega-electron-volt ultrafast electron diffractionREVIEWER COMMENTS

Reviewer #1 (Remarks to the Author):

The authors reported an ultrafast MeV-electron diffraction (UED) study of VO₂. While this system has long been studied by versatile experimental and theoretical techniques to elucidate the fascinating character of its photoinduced insulator-to-metal transition, there still remains a strong controversy over fundamental mechanisms. Owing to the fabrication of quasi-single-crystal films, the data quality is higher than previous UED studies, where polycrystalline samples only provided radially-integrated 1D profile and prevented from clear understanding.

However, although the results are in principle intriguing and most parts of the manuscript are convincing, I have a significant concern that needs to be addressed.

(1) The largest concern is an interpretation of the peak width change of the 200R (integrated over 002R/200R/202 \bar{R} spots from three domains and their twinning) spot along the azimuthal direction. Although it is clear that the 200R width decreases with a different slow time scale from the initial fast evolution associated with breaking of V-V dimers and zigzag structure, the attribution of the peak narrowing should be carefully performed because 6 spots (3 domains and 2 twinning) contribute to the formation of this unresolved diffraction spot. The authors attributed the observed peak narrowing to suppression of twinning. But I wonder how authors could exclude other possibilities, e.g., it may arise from the narrowed peak separation between 002R/200R diffraction spots arising from different 2 domains.

Indeed, in Fig. 1d and Fig. S2, the 3 diffraction spots (002R/200R/202 \bar{R}) arising from 3 domains are more separated azimuthally than the separation arising from twinning, in simulation. Although negative transient signal can be observed in Fig. 2a along the azimuthal direction (it seems corresponding to 002R/200R spots in my eyes), I am not sure whether this can be also observed along another azimuthal direction corresponding to the 202 \bar{R} spot. If the observed decrease in the peak width is attributed only to the suppress of twinning, the magnitude in the peak width decrease is expected to be relatively more significant along 202 \bar{R} spot than along 002R/200R spots. This expectation seems natural as the broadening effect contributed from different 2 domains is smaller along the 202 \bar{R} spot than along 002R/200R spots. However, in Fig. 2a, it looks to me that the observed signal is not following this expectation. Furthermore, in ref. (37), single-crystal VO₂ does show the peak intensity change but no width change. Thus, in the current description, I can not help thinking the possibility that the peak width decreases because the relative orientation between three 120 degree rotated domains could be modulated due to thermal effect, strain gradients, and other reasons.

To unambiguously attribute the decrease in the peak width to suppression of twinning, quantitative evaluation is definitely needed.

(1-1) What is the intrinsic separation of diffraction spots arising from twinning?

(1-2) Can the suppression of the twinning quantitatively explain the observed decrease in the peak width?

(1-3) Why other possible scenarios can be excluded? (Particularly, how can authors fix the separation between Bragg spots arising from 3 domains as constant?)

(1-4) What the differences from the ref. (37) make it possible to observe the suppress of twinning in this experiment? Is there any improvement in the momentum or azimuthal resolutions?

The (2)-(3) below are minor comments.

(2) There is no (vertical) unit, numbers in the inset of Figure 2d, thus it is hard to follow the relative peak width decrease as a function of delay time.

(3) In Figure 2c and 2d, based on the thermal transition data (Fig. S3), it is better to show (or write in the inset) the level of relative change when 100 % M1-R transition occurs in the probed volume.

Reviewer #2 (Remarks to the Author):

Review of the manuscript

"Transient monoclinic structure without vanadium dimers and zigzag chains in quasi-single-crystal VO₂ revealed by mega-electron-volt ultrafast electron diffraction"

This work aims to study the crystallographic evolution of VO₂ by ultrafast electronic diffraction, in particular the transient structures related to the position of the vanadium atoms. The chosen technique is powerful and well suited to the objective.

However, the manuscript has two major flaws:

- the structure and the microstructure of the sample (film of VO₂) were not sufficiently characterized before the experiments.
- the data analysis lacks rigour, which leads to potentially erroneous crystallographic interpretations.

The authors need to work on the issues mentioned above and outlined below.

Manuscript

58. intermediate phase without dimers but with zigzag chains. ("with" is missing).

64-65: "However, so far, no information about crystal symmetry transformation has been obtained in diffraction experiments".

This sentence contradicts the information on the V given previously. Authors should specify what they mean by "crystal symmetry".

83. "we succeed in fabricating high quality quasi-single crystalline VO₂ freestanding films".

The structural characterization of the VO₂ films is not sufficient. The notion of quasi-crystallinity is not clear. The AFM image (Fig S1) seems to show a polycrystalline film instead. It is essential to precisely characterize the micro/nanostructure of the layer before interpreting the diffracted intensities. The AFM image in the supplementary material must be explained (topography, grain size distribution, orientation relations, possible texture, ...). The roughness information of the layer must be based on a profile curve. The text mentions three domains: the authors should indicate them on the AFM image. What is their relative proportion?

110-112. "Miller indexes with the "R" subscript correspond to the non-superstructure spots both in the M1 and the R phase."

This choice of indexing is detrimental to the understanding of the pattern. If the indexing is based on the monoclinic unit cell, the hkl index must be M and not R.

112-113. "Owing to the in-plane domains, there are 12 equivalent groups of the $[102]_M$ / $[302]_M$ spots marked by white circles."

The white circles are not sufficiently visible. What does "equivalent" mean? What is the indexing of other reflection groups included in the white circles. In particular, is each elongated spot of one group equivalent in intensity to the elongated spot with same indices (but coming from another domain) in another group?

113-114. "There are six equivalent groups of $[002]_R$ / $[200]_R$ / $[202]_R$ spots and six equivalent groups of $[202]_R$ / $[2\bar{0}4]_R$ spots."

Same remark.

116. "Small dots represent diffraction spots due to V-V dimers in M1 phase."

The sentence is confusing.

139. "The static electron diffraction pattern at 300 K measured by 200 keV TEM is shown in Fig. 1(c) with Miller indexes shown for several representative diffraction spots."

The authors should give the acquisition conditions for the diffraction pattern: parallel beam, use of

selected area aperture, size of the diffracting region, etc...

140-142. "Superstructure spots in the M1 phase are marked with a "M" subscript; non-superstructure spots existing in both M1 and R phase are marked with a "R" subscript. All Miller indexes use M1-phase's representation."

This choice of indexing is detrimental to the understanding of the pattern. If the indexing is based on the monoclinic unit cell, the hkl index must be M and not R.

145-147. "Both the pattern and the elongated shape of the diffraction spots from VO₂ can be understood based on three 120° in-plane domains (plus twinning in each domain) and the anisotropic shape of the crystal grain."

The argument is not convincing. The authors should dissociate the explanation related to the presence of domains from that on the shape of the grains. A profile of the lines along the direction of elongation would help.

151. "two sets of sports spots because of twin domains"

152. "For example, labelled by the magenta square in Fig. 1(d), 002_R, 200_R, 202_R spots are three equivalent spots from three domains."

How can these reflections be equivalent when they are obviously not at the same distance from the transmitted beam?

164. "As expected, the pure superstructure spots $[102]_{-M}$ completely disappear above the transition temperature (Fig. 1(e)-right)."

However, it seems that there is still some intensity at the location of 102M on figure 1e. An intensity profile would help.

167-168. "The superstructure peaks ($[100]_{-M}$, $[1\bar{0}2]_{-M}$ and $[102]_{-M}$) are only a small fraction of the background."

The sentence is confusing.

170. "Following the temperature dependence of the intensity of $[102]_{-M}$ peak"

Insofar as this "peak 102M" actually brings together the diffracted intensities of 102 and 30-2, each coming from a different domain, what is the relevance of studying the evolution of its intensity?

178-179. "Time evolution of the intensity for (b) 102M peaks and (c) 200R peaks. (d) Time evolution of the width of 200R peaks. "

What is the relevance of studying the total intensity of the group 200, 002 and 20-2, which come, according to the authors, from 3 domains of different orientations.

187. "Blue dot indicates the diffraction spot from the superstructure due to V-V dimers."

What justifies the extinction of 102 in the situation presented in Figure 2f? What is the structure factor?

189-302.

Is it reasonable to interpret these intensity variations, which are actually the sum of the intensities of reflections of different indices and coming from different grains? Does the phase transition occur uniformly throughout the diffracting domain, regardless of grain size or orientation? Does the position of the grain in the thickness of the layer have an influence?

214. "diffraction spots" (instead of sports)

287. "found this new process in quasi-single (instead of signal) crystalline samples"

315. "roughness (instead of Thickness) of VO₂ film were checked by AFM. "

How was VO₂ film thickness measured?

Supplementary material

1) Crystal structure and morphology of VO₂ films.

"The epitaxial relationship is..."

An X-ray diffraction experiment, for example, would highlight the epitaxy relationship between VO₂ and STO.

"STO (111) has 6-fold symmetry"

3-fold axis

2) Simulation on the TEM diffraction pattern of VO₂ film

Figure S2c : Indexation de la phase R dans le réseau monoclinique.

Please indicate the zone axis [010] in the figure. The actual indexing corresponds to the monoclinic form. To make the simulation easier to read, please indicate M (and not R) as the hkl index. Indexing some reflections in the 2 systems would help (example 002M and 200R, 40-2M and 002R).

Figure S2d :

Monoclinic, so hkIM.

"M1-phase has superstructure spots (small spots in Fig. S2(d)) from the V-V dimers. "

The sentence is confusing.

4) Lattice dynamics after..

Fig S4 b

The -102M reflection is almost superimposed with an STO reflection. How was the STO contribution withdrawn?

5) The relation between the peak width of 200R peaks and the twin domains

Fig S5 c, d, e

The peak width difference (c and e) is 1 pixel. Is it significant? What is the measurement uncertainty?

Reviewer #3 (Remarks to the Author):

This paper reports an important investigation of atomic dynamics during the photo-induced phase transition (PIPT) in quasi-single-crystal VO₂ using ultrafast electron diffraction (UED) with 50 femtosecond temporal resolution and a 3 MeV electron source. Its major conclusion is that the PIPT first drives a transient monoclinic phase in which the vanadium ion chains are neither dimerized nor in a zig-zag chain configuration before settling into the quasi-equilibrium R phase about 5 ps after the initial photoexcitation event. The paper is publishable after one major and a few minor weak points are corrected.

Strengths: The growth technique that produces the free-standing quasi-single-crystal films is unusual and provides a well-defined growth axis that enables an unambiguous test of the way in which the vanadium ion chains evolve following photoexcitation. The analysis of the MeV-UED results is careful and supports the overall conclusion; at the same time, it clarifies the role of coherently excited phonons and provides plausible support for a 2016 paper that proposed that the PIPT proceeds from the M1 phase through M2 to the final R phase. The literature review is thorough and the interpretation of the meaning of previous results in the context of the present results is generally well argued.

Weaknesses, corrections and suggestions: The major weakness in the paper is the repeated assertion that the new measurements do not support the existence of the monoclinic metallic (mM) state reported in References 18, 32 and 35. Because the present work presents only diffraction data, the only basis for this assertion seems to be the duration of the monoclinic transient and its lack of a second excitation threshold. However, diffraction measurements alone

do not yield measurements of free-electron density that confirm metallicity. (See, for example, the side-by-side measurements of diffracted intensity and THz conductivity presented in Fig 4 of Ref. 35.). Thus references to the mM phase (for which there is solid evidence in LEEM experiments) should be omitted.

Because the paper makes comparisons both to bulk single-crystal and polycrystalline thin-film data, it would be helpful if the authors would spell out precisely what they mean by "quasi-single-crystal." It sounds like all that is intended is that their samples are well lattice-matched with some twinning, large domain sizes and a clearly defined film plane as suggested by the TEM simulation in Fig. 1(d); if that is indeed the case they should also give the lattice parameters and matching statistics. It would also be useful to state the lateral dimensions in the main text.

In commenting on Ref. 37 (page 3) the authors wrongly characterize the VO₂ sample as having been "fabricated by focused ion beam" (FIB), when in fact the sample is a bulk single crystal that has been thinned by FIB.

Style and mechanics issues: The title is so long that it almost seems like an abbreviated abstract, and yet does not include the essential information that the transient monoclinic phase is generated by photoexcitation! I suggest something like "Transient dynamics of the phase transition in VO₂ by ultrafast 3 MeV electron diffraction."

The paragraphs on pages 8 and 9 of the manuscript are so long that the thread of analysis is difficult to follow. The authors need to break up these paragraphs so that each new point in the discussion is allotted a separate paragraph.

In Figure 1(a), it is difficult to see the black type against the dark colors of the layers in the sample; indeed, the SAO is completely invisible. The type should be yellow or white.

Apart from occasional lapses in wording that indicate that the authors are not native English speakers, there are many misspellings, some of which actually muddle the intended meaning. It would be advisable to have the manuscript proofread by a native English speaker.

We want to thank the referees for their very careful review and constructive suggestions to help us improve our manuscript.

Here is a list of main changes in the revised manuscript:

1. Following the reviewer's suggestion, we carried out more quantitative analysis on the origin of the evolution of peak width. Fig. S5 in supplementary materials was revised to include the new analysis. More discussions were added in the main text.
2. Following the reviewer's suggestion, the structure of our samples was sufficiently characterized. Fig. S1 in supplementary materials was revised to include the new data. More discussions were added in the main text.
3. Following the reviewer's suggestion, we revised the title to "Transient dynamics of the phase transition in quasi-single-crystal VO₂ by MeV ultrafast electron diffraction"
4. For easy reading, we reorganized the figures. New Fig. 1 consists of the previous Fig. 1(a)-(d); New Fig. 2 consists of the previous Figs. 1(e), (f) and the previous Figs. (a)-(d). A new panel (Fig. 2(b)) was added in new Fig. 2 according to referee's suggestion; New Fig. 4 consists of the previous Figs. 2(e)-(g); New Fig. 5 is the previous Fig. 3.

We highlight the revised text in the manuscript.

Below are our point-to-point responses to reviewers' comments on our manuscript.

Reviewer #1:

The authors reported an ultrafast MeV-electron diffraction (UED) study of VO₂. While this system has long been studied by versatile experimental and theoretical techniques to elucidate the fascinating character of its photoinduced insulator-to-metal transition, there still remains a strong controversy over fundamental mechanisms. Owing to the fabrication of quasi-single-crystal films, the data quality is higher than previous UED studies, where polycrystalline samples only provided radially-integrated 1D profile and prevented from clear understanding.

However, although the results are in principle intriguing and most parts of the manuscript are convincing, I have a significant concern that needs to be addressed.

(1) The largest concern is an interpretation of the peak width change of the 200R (integrated over 002R/200R/20 $\bar{2}$ R spots from three domains and their twinning) spot along the azimuthal direction. Although it is clear that the 200R width decreases with a different slow time scale from the initial fast evolution associated with breaking of V-V dimers and zigzag structure, the attribution of the peak narrowing should be carefully performed because 6 spots (3 domains and 2 twinning) contribute to the formation of this unresolved diffraction spot. The authors attributed the observed peak narrowing to suppression of twinning. But I wonder how authors could exclude other possibilities, e.g., it may arise from the narrowed peak separation between 002R/200R diffraction spots arising from different 2 domains.

Reply: We want to thank the referee for helping us improve the clarity of our manuscript. **It should be noted that we reindex the 002R/200R/20 $\bar{2}$ R as 00 $\bar{2}$ /200/ $\bar{2}$ 02 to exactly match the relative orientation of three 120° domains in the revised manuscript.** The referee is right that one spot does include 6 spots (from 3 domains and twinning) in MeV UED. When VO₂ changes from the M₁ phase to R phase in our films, the suppression of twinning domains and the reduction of the peak separation between 00 $\bar{2}$ and 200 occur simultaneously. They both are the consequences of the structural transition from monoclinic to tetragonal structure. In our previous analysis, for simplicity, we assumed that twin domains play the dominant role in the narrowing of

peak width. Following the suggestion of the referee, we carried out more quantitative analysis and found out that suppression of twinning domains and the narrowed peak separation between $00\bar{2}$ and 200 have comparable contributions to the narrowing of the overall peak width. We presented the details in the reply to Comment #1-2 below and clarified this issue in the revised version. The new analysis does not change our findings that the disappearance of vanadium dimers and zigzag chains does not coincide with the transformation of crystal symmetry.

(2) Indeed, in Fig. 1d and Fig. S2, the 3 diffraction spots ($002R/200R/20\bar{2}R$) arising from 3 domains are more separated azimuthally than the separation arising from twinning, in simulation. Although negative transient signal can be observed in Fig. 2a along the azimuthal direction (it seems corresponding to $002R/200R$ spots in my eyes), I am not sure whether this can be also observed along another azimuthal direction corresponding to the $20\bar{2}R$ spot. If the observed decrease in the peak width is attributed only to the suppress of twinning, the magnitude in the peak width decrease is expected to be relatively more significant along $20\bar{2}R$ spot than along $002R/200R$ spots. This expectation seems natural as the broadening effect contributed from different 2 domains is smaller along the $20\bar{2}R$ spot than along $002R/200R$ spots. However, in Fig. 2a, it looks to me that the observed signal is not following this expectation. Furthermore, in ref. (37), single-crystal VO₂ does show the peak intensity change but no width change. Thus, in the current description, I cannot help thinking the possibility that the peak width decreases because the relative orientation between three 120° rotated domains could be modulated due to thermal effect, strain gradients, and other reasons.

Reply: In our previous version, we lacked of the description about the green line in previous Fig. 1(e) (now it is Fig. 2(a)). We clarified this issue in the revised version. In fact, the green line does not indicate the momentum position where we get the line profile. It only marks a direction. In order to obtain the line profile of the diffraction

peak with high signal-to-noise ratio along the green line (called x-direction in the revised manuscript), we superposed the intensity of ($\bar{2}02/00\bar{2}/200$) spot along the direction perpendicular to the green line (called y-direction in the revised manuscript). If we only use a single cut (slice) of data without superposition, for example exactly along the position corresponding to $\bar{2}02$ spot as reviewer suggested, the signal-to-noise ratio is not good enough to carry out peak fitting. We carried out more analysis to show how $\bar{2}02$ spot affects the peak width in the revised version.

Fig. R1 The effect of $\bar{2}02$ spot. (a) Time evolution of the width of ($\bar{2}02/00\bar{2}/200$) spot along x-direction and y-direction. Simulated line profiles of $00\bar{2}$ (green), 200 (red), $\bar{2}02$ (black) spots and their superposition (blue) along (b) x-direction and (c) y-direction.

Fig. R1(a) shows the evolution of the peak width (FWHM) along x-direction and y-direction. The evolution of peak width is anisotropic. Peak width along y-direction is

nearly constant. This anisotropic behavior can be understood by Figs. R1(b) and (c). Shown in the inset of Fig. R1(a), there are six spots, i.e. $\bar{2}02$, $00\bar{2}$, 200 and their twins. Figs. R1(b) and (c) present the line profiles of these six spots along x-direction and y-direction, respectively. The peak height is set according to the structure factor and peak width is set to be the same for simplicity. It should be noted that the intensity of $\bar{2}02$ spot is about 1/6 of $00\bar{2}/200$ spots according to the structure factor. To compare with our data, we need to add up all the lines to get the final line profile (blue line).

For x-direction, shown in Fig. R1(b), both the twin domains splitting and the separation between $00\bar{2}$ and 002 will contribute to the change of peak width along x-direction. Therefore, we can observe peak width narrowing along the x-direction when VO_2 transforms from the M_1 to R phase.

For y-direction, it is very different. The separation between $00\bar{2}$ and 200 , as well as the twin domain splitting of $\bar{2}02$ peak does not contribute to the change of peak width along y-direction, because such separation and splitting are along x-direction. The y-component of the twin domain splitting of $00\bar{2}$ and 200 spots can contribute to the change of peak width, however it is very small. On the other hand, $\bar{2}02$ spot will move closer to $00\bar{2}/200$ spots along y-direction during the transition from the M_1 to R phase, which could reduce the peak width in principle. However, $\bar{2}02$ peak is much weaker than $00\bar{2}/200$ peak and can only contribute to the left tail of the overall line profile. Therefore, when the line profile is fitted by a Gaussian peak, the obtained peak width is nearly unaffected by the movement of $\bar{2}02$ peak. As a result, the change of the peak width along the y-direction should be much weaker than that along the x-direction.

We present detailed discussion and evaluation in the revised supplementary materials Fig. S5. Some calculations are also presented in the reply to Comment #1-2 below.

Because of the experimental condition, Ref. 37 cannot observe the narrowing of peak width. We present the reason in the reply to Comment #1-4.

Comment #1-1: What is the intrinsic separation of diffraction spots arising from twinning?

Reply: The intrinsic separation arising from twinning is about 0.004 \AA^{-1} . More details are shown in the reply to Comment #1-2.

Comment #1-2: Can the suppression of the twinning quantitatively explain the observed decrease in the peak width?

Fig. R2 200, $00\bar{2}$ and $\bar{2}02$ spots in (a) M_1 phase and (b) R phase.

Reply: As we discussed above, both twinning and 120° domains contribute to the peak width along x-direction. Marked in Fig. R2(a), the splitting of $\bar{2}02$ spot arising from twinning in the M_1 phase is $\sim 0.004 \text{ \AA}^{-1}$ along x-direction. The twin domain splitting of 200 or $00\bar{2}$ spots is $\sim 0.001 \text{ \AA}^{-1}$ along x-direction. The separation between 200 and $00\bar{2}$ spots is $\sim 0.031 \text{ \AA}^{-1}$ in the M_1 phase and $\sim 0.026 \text{ \AA}^{-1}$ in the R phase. It changes by $\sim 0.005 \text{ \AA}^{-1}$. We can roughly estimate the largest relative change of peak width after transition from the M_1 to R phase. If we assume the total peak width in the M_1 phase is $\sim 0.031 \text{ \AA}^{-1}$ and take 0.005 \AA^{-1} as the change of width, then we get the largest relative change of peak width is about -16%. In real experiments, the total peak width of the diffraction spots becomes much larger because of the crystal quality and beam size, the relative change of the peak width must be much less than -16%. In fact, the total peak width is about 0.07 \AA^{-1} in our samples, corresponding to a relative change about -7%. In our experiments, the extracted relative change of peak width is about -5%, which is

consistent with the explanation. More evaluations are shown in the revised Fig. S5 in SM.

Comment #1-3: Why other possible scenarios can be excluded? (Particularly, how can authors fix the separation between Bragg spots arising from 3 domains as constant?)

Reply: Referee is right, we cannot fix the separation. As we discussed above, in the revised version, we found that both the twin domains and the 120° domains would lead to the decrease of the peak width during the structural transition.

Comment #1-4: What the differences from the ref. [37] make it possible to observe the suppress of twinning in this experiment? Is there any improvement in the momentum or azimuthal resolutions?

Reply: There are two main differences between ref. [37] and our work. First, effective pump laser fluence in ref. [37] is too low. Second, they do not have 120° domains and the ratio of twin domains is also low.

In ref. [37], the M_1 phase can completely change to the R phase by heating. Fig. R3(a) and (b) show diffraction patterns for the M_1 phase and the R phase in thermal equilibrium from ref. [37]. In the M_1 phase (Fig. R3(a)), there are weak superstructure peaks, such as $10-2_{M_1}$, originating from the V-V dimers. Those superstructure peaks totally disappear in the R phase (Fig. R3(b)). Consequently, the index of the diffraction spots also changes. For example, $20-2_{M_1}$ spot become 101_R and $40-2_{M_1}$ become 002_R . It must be noted that the difference between the M_1 and the R phase is not only the superstructure peaks. The symmetry of the lattice is also different if we focus on the primary diffraction peaks. To show this, we draw two similar cells based on the primary spots in Fig. R3(a) (blue dashed lines) and R3(b) (magenta dashed lines). In the M_1 phase, we obtain a **parallelogram** cell. In the R phase, the cell is a **rhombus**. The difference between two cells is clear when we overlaid them in Fig. R3(c).

Editorial Note: Figure below modified from Li, J. et al. Direct Detection of V-V Atom Dimerization and Rotation Dynamic Pathways upon Ultrafast Photoexcitation in VO₂. *Phys. Rev. X* **12**, 021032, under a CC BY 4.0 licence

Fig. R3 Static electron diffraction pattern in (a) M₁ phase and (b) R phase from ref. [37]. (c) Overlay of the cells from (a) and (b). Time-resolved diffraction patterns at (d) t = 0, (e) t = 9 ps from ref. [37]. (f) Overlay of the cells from (d) and (e).

However, in the time-resolved experiments in ref. [37], the M₁ phase is only slightly suppressed. Figs. R3(c) and (d) show the time-resolved diffraction patterns at t = 0 and t = 9 ps after pumping, respectively from ref. [37]. The superstructure peaks are only slightly suppressed. According to ref. [37], then intensity of superstructure peaks only decreases by about 20% after pumping. Again, we draw two cells and overlaid them in Fig. R3(f). They completely overlap indicating the change of symmetry is still undetectable. In fact, calculations in ref. [37] show that it is in the very initial stage of the M₁ to R phase transition. The effective laser fluence in ref. [37] is too low to produce notable change of peak width. Furthermore, the absence of 120° domains and the small

ratio of twin domains also make it difficult to trace the change of peak width under their experimental conditions in ref. [37].

Comment #2: There is no (vertical) unit, numbers in the inset of Figure 2d, thus it is hard to follow the relative peak width decrease as a function of delay time.

Reply: We want to thanks the referee for a very careful review. We added the vertical unit and numbers in the revised version.

Comment #3: In Figure 2c and 2d, based on the thermal transition data (Fig. S3), it is better to show (or write in the inset) the level of relative change when 100 % M1-R transition occurs in the probed volume.

Reply: We want to thanks the referee for helping us improve the clarity of our manuscript. We marked the level of relative changes (black dashed lines) when 100% M₁-R transition occurs based on the thermal transition data in the insets of Fig. 2(f) and (g) (Fig. 2(c) and (d) in the previous version).

Reviewer #2

This work aims to study the crystallographic evolution of VO₂ by ultrafast electronic diffraction, in particular the transient structures related to the position of the vanadium atoms. The chosen technique is powerful and well suited to the objective.

However, the manuscript has two major flaws:

- the structure and the microstructure of the sample (film of VO₂) were not sufficiently characterized before the experiments.
- the data analysis lacks rigour, which leads to potentially erroneous crystallographic interpretations.

The authors need to work on the issues mentioned above and outlined below.

We want to thank the referee for the constructive suggestion that have greatly improved the quality and clarify of the manuscript. In the revised version, more structural characterization and more quantitative data analysis were given.

Comment #1:

58. intermediate phase without dimers but with zigzag chains. (“with” is missing).

Reply: We corrected in the revised version.

Comment #2:

64-65: “However, so far, no information about crystal symmetry transformation has been obtained in diffraction experiments”. This sentence contradicts the information on the VO₂ given previously. Authors should specify what they mean by “crystal symmetry”.

Reply: We want to thank the referee for the suggestion. Transition from the M₁ to the R phase involves not only the change of fractional coordinates of V atoms within a unit cell, but also the change of the lattice constants a_M and c_M , as well as the angle β between them. Information of a_M , c_M and β is shown in Fig. 3. The crystal is monoclinic in the M₁ phase and becomes tetragonal in the R phase. “crystal symmetry”

means monocline or tetragon. While, previous reports only focused on the fractional position of V atoms within a unit cell, the question whether the change of the fractional position of V atoms coincides with the change of the crystal symmetry remains unanswered. In the revised manuscript, we explained the meaning of “crystal symmetry” in the main text in page 2, last paragraph.

Comment #3:

83. “we succeed in fabricating high quality quasi-single crystalline VO₂ freestanding films”. The structural characterization of the VO₂ films is not sufficient. The notion of quasi-crystallinity is not clear. The AFM image (Fig S1) seems to show a polycrystalline film instead. It is essential to precisely characterize the micro/nanostructure of the layer before interpreting the diffracted intensities. The AFM image in the supplementary material must be explained (topography, grain size distribution, orientation relations, possible texture, ...). The roughness information of the layer must be based on a profile curve. The text mentions three domains: the authors should indicate them on the AFM image. What is their relative proportion?

Reply: Thanks for the referee’s valuable advice. In the revised manuscript, we added more structural characterization of the VO₂ films. Our VO₂ film is not a polycrystal. As shown in Fig. S1(f) in the supplementary materials (SM), There are lots of crystal grains, but only three in-plane orientations with an interval of 120° (called 120° domains for convenience) were observed. All the grains have the same out-of-plane orientation, which is [010]. Within the TEM (4 μm in diameter) or UED (150 μm in diameter) probing region, three domains are equally distributed. In the revised Fig. S1 in SM, detailed information following the referee’s suggestions were presented. In page 4, 2nd paragraph of the revised main manuscript, we gave the clear meaning of “quasi-single crystalline”.

Comment #4:

110-112. “Miller indexes with the “R” subscript correspond to the non-superstructure

spots both in the M1 and the R phase.” This choice of indexing is detrimental to the understanding of the pattern. If the indexing is based on the monoclinic unit cell, the hkl index must be M and not R.

Reply: We corrected the index in the revised version.

Comment #5:

112-113. “Owing to the in-plane domains, there are 12 equivalent groups of the 102M/30-2M spots marked by white circles.” The white circles are not sufficiently visible. What does "equivalent" mean? What is the indexing of other reflection groups included in the white circles? In particular, is each elongated spot of one group equivalent in intensity to the elongated spot with same indices (but coming from another domain) in another group?

Reply: In the revised Fig. 1(c), we increased the thickness of the white circle to make it more visible. We removed subscript “M” in the revised version.

Following the referee’s suggestion, indices of all the spots within the blue circles (they are the same as the white circles in Fig. 1(c)) were added in the revised Fig. 1(d). There are two spots in each group. The inner 12 spots are $102/102_T/\bar{1}0\bar{2}/\bar{1}0\bar{2}_T$. Each 120° domain contributes four spots when twinning is considered. The subscript “T” means the contribution from a twin domain. 102 and 102_T are essentially equivalent. Since VO_2 has the inversion symmetry, 102 and $\bar{1}0\bar{2}$ are equivalent. So, 102 , 102_T , $\bar{1}0\bar{2}$ and $\bar{1}0\bar{2}_T$ are indeed equivalent spots. The outer 12 spots are $30\bar{2}/30\bar{2}_T/\bar{3}0\bar{2}/\bar{3}0\bar{2}_T$. They are also equivalent spots. Therefore, such 12 groups are equivalent.

The referee is right that each elongated spot of one group equivalents in intensity to the elongated spot with same or equivalent indices (but coming from another domain) in another group. We clarified the meaning of “equivalent” in page 5, last paragraph of the revised manuscript.

Comment #6:

113-114. “There are six equivalent groups of 002R/200R/202R spots and six equivalent

groups of 202R/-204R spots.” Same remark

Reply: The same explanation as the reply to Comment 5. More indices were added in the revised Fig. S2(f) in SM to show that six groups have spots with the equivalent or same indices but from different domains. We clarified this issue in page 5, last paragraph of the revised manuscript.

Comment #7:

116. “Small dots represent diffraction spots due to V-V dimers in M1 phase.”

The sentence is confusing.

Reply: In the revised manuscript, we changed this sentence to “For convenience, we use smaller dots to present the superstructure spots.” in the caption of Fig. 1. It should be noted, we defined the meaning of “superstructure” in page 2, 1st paragraph in the main text, as following: “In the monoclinic insulating phase (called the M₁ phase), V atoms dimerize and form zigzag chains (called the superstructure in this work)”

Comment #8:

139. “The static electron diffraction pattern at 300 K measured by 200 keV TEM is shown in Fig. 1(c) with Miller indexes shown for several representative diffraction spots.” The authors should give the acquisition conditions for the diffraction pattern: parallel beam, use of selected area aperture, size of the diffracting region, etc...

Reply: The incident electron beam is along [010] direction of VO₂. The selected area aperture is 200 μm in diameter and the size of the diffracting region is 4 μm in diameter. We added the acquisition conditions in the section of “Method”. The incident beam direction is indicated in the caption of Fig. 1.

Comment #9:

140-142. “Superstructure spots in the M1 phase are marked with a “M” subscript; non-superstructure spots existing in both M1 and R phase are marked with a “R” subscript. All Miller indexes use M1-phase’s representation.” This choice of indexing is

detrimental to the understanding of the pattern. If the indexing is based on the monoclinic unit cell, the hkl index must be M and not R.

Reply: We corrected the index in the revised manuscript. All Miller indices use the M_1 -phase's representation. Subscript of "M" and "R" was removed.

Comment #10:

145-147. "Both the pattern and the elongated shape of the diffraction spots from VO₂ can be understood based on three 120° in-plane domains (plus twinning in each domain) and the anisotropic shape of the crystal grain." The argument is not convincing. The authors should dissociate the explanation related to the presence of domains from that on the shape of the grains. A profile of the lines along the direction of elongation would help.

Reply: Thanks for the referee's instructive suggestions. We removed this sentence and added more discussion to clarify this issue. The elongated shape along the tangential direction in Fig. 1(c) is related to two effects. First, three sets of diffraction spots from 120° in-plane domains distribute in the tangential direction, seen from Fig. 1(d). They partially overlapped in experiments, which makes the spots broaden along the tangential direction. Second, the in-plane shape of crystal grains is anisotropic, which also makes the spots broader in the tangential direction than in the radial direction. In the revised Fig. S1 in SM, we added more structural characterization including high resolution AFM topography to show the shape of the grains. The line profiles of diffraction spots along two directions were also plotted in Fig. S1. We also clarified this issue in page 4, last paragraph in the revised manuscript.

Comment #11:

151. "two sets of sports (spots) because of twin domains"

Reply: We corrected in the revised manuscript.

Comment #12:

152. “For example, labelled by the magenta square in Fig. 1(d), 002R, 200R, 20-2R spots are three equivalent spots from three domains.” How can these reflections be equivalent when they are obviously not at the same distance from the transmitted beam?

Fig. R4 Demonstration of three 120° domains

Reply: It should be noted that we reindexed the $002R/200R/20\bar{2}R$ as $00\bar{2}/200/\bar{2}02$ to exactly match the relative orientation of three 120° domains in the revised manuscript.

The referee is right. Our previous description is not accurate. Fig. R4 demonstrates $00\bar{2}$, 200 and $\bar{2}02$ spots coming from three 120° domains. Blue, red and black parallelograms mark the unit cells in the reciprocal space from three 120° domains. The unit cell is not a rhombus, so $00\bar{2}$, 200 and $\bar{2}02$ spots are not at the same distance from the transmitted beam. Therefore, we cannot call them equivalent diffraction spots. However, these three spots have a common property. When VO_2 changes from the M_1 phase to R phase, the intensities of $00\bar{2}$, 200 and $\bar{2}02$ spots are all enhanced, which were observed and analyzed in the diffraction experiments on single crystalline samples (ref. 37 in our manuscript). In this sense, we call them “equivalent”. To avoid ambiguity, we removed “equivalent” in the revised manuscript. We clarified that the intensity of these three spots were all enhanced during the phase transition in page 7, 2nd paragraph in the revised manuscript.

Comment #13:

164. “As expected, the pure superstructure spots 102M completely disappear above the transition temperature (Fig. 1(e)-right).” However, it seems that there is still some intensity at the location of 102M on figure 1e. An intensity profile would help.

Reply: We want to thank the referee for the constructive suggestion. An intensity profile was added in the revised Fig. 2(b). As seen from the line profiles, the diffraction peak is well-defined at 300 K. At 345 K, the diffraction peak disappears and the background is slightly enhanced due to the thermal effect.

Comment #14:

167-168. “The superstructure peaks (100M, -102M and 102M) are only a small fraction of the background.” The sentence is confusing.

Reply: What we want to say is that these superstructure peaks are influenced by the non-superstructure peaks with very high intensity. Signal-to-noise ratio of the superstructure peaks in polycrystalline samples is relatively low, so superstructure peaks cannot be well resolved. We changed this sentence to “The superstructure peaks (100, $\bar{1}02$, and 102) are influenced by the neighboring non-superstructure peak (200) with very high intensity, which renders accurate determination of the threshold difficult in polycrystalline samples”

Comment #15:

170. “Following the temperature dependence of the intensity of 102M peak” Insofar as this “peak 102M” actually brings together the diffracted intensities of 102 and 30-2, each coming from a different domain, what is the relevance of studying the evolution of its intensity?

Reply: The referee is right that 102 and $\bar{3}02$ ($30\bar{2}$ is reindexed as $\bar{3}02$ in the revised manuscript) come from two different domains. Although they have different index, 102 and $\bar{3}02$ behave similarly when VO_2 changes from the M_1 to the R phase. 102 and $\bar{3}02$ spots are only existed in the M_1 phase related to the V-V dimers. They are

superstructure spots. The structure factor of 102 and $\bar{3}02$ are zero in the R phase, so they both disappear during the SPT from the M_1 phase to R phase. In our UED experiment, the diameter of pump beam is 1.5 mm and the diameter of electron beam is 150 μm . As discussed in Fig. S1 in SM, the volume fractions of domains are equal within our pump and probe region. Therefore, what we obtained in our experiments is the overall intensity changes of (102/ $\bar{3}02$) as a function of time after pumping. Although we cannot obtain the individual time constant of 102 or $\bar{3}02$ peak, the experimental determined timescale of 200 fs based on the disappearance of (102/ $\bar{3}02$) peak gives the time constant when V-V dimers melt and V atoms reach the same fractional coordinates as that in the R phase. We clarified why we can study the dynamics even if 102 and $\bar{3}02$ spots overlap in page 7, 2nd paragraph in the revised manuscript.

Comment #16:

178-179. “Time evolution of the intensity for (b) 102M peaks and (c) 200R peaks. (d) Time evolution of the width of 200R peaks. “What is the relevance of studying the total intensity of the group 200, 002 and 20-2, which come, according to the authors, from 3 domains of different orientations.

Reply: We reindex the 002/200/20 $\bar{2}$ as 00 $\bar{2}$ /200/ $\bar{2}02$ in the revised manuscript. When VO_2 changes from the M_1 to R phase, 00 $\bar{2}$, 200, and $\bar{2}02$ spots behave similarly. In previous UED experiments on polycrystalline samples (Ref. 18, 19, 35, 38), 200 peak they studied consist of 00 $\bar{2}$, 200, 002, and $\bar{2}00$. $\bar{2}02$ peak was not resolved because its low intensity. Revealed in previous UED study on a single crystal VO_2 (ref. 37), 00 $\bar{2}$, 200, and $\bar{2}02$ spots were all enhanced after photoexcitation. It should be noted that our 00 $\bar{2}$ /200/ $\bar{2}02$ correspond to $\bar{2}02$ /200/00 $\bar{2}$ in Ref. 37 because they choose a different unit cell. In addition, these three spots have very similar time constant according to ref. 37. Therefore, we think it is reasonable to study the evolution of the total intensity of the group 00 $\bar{2}$, 200, and $\bar{2}02$.

Comment #17:

187. “Blue dot indicates the diffraction spot from the superstructure due to V-V dimers. What justifies the extinction of 102 in the situation presented in Figure 2f? What is the structure factor?”

Reply: When VO₂ is in the R phase, structure factor of 102 is zero. When it is in the M1 phase, V atoms form dimers. Correspondingly, we will observe superstructure spots, such as 102 spot. After photoexcitation, those dimers will be destroyed and 102 spot gradually disappear. Fig. 3(b) (previous Fig. 2f) shows the situation where dimers are completely destroyed. In this situation, the structure factor of 102 is zero.

Comment #18:

189-302. Is it reasonable to interpret these intensity variations, which are actually the sum of the intensities of reflections of different indices and coming from different grains? Does the phase transition occur uniformly throughout the diffracting domain, regardless of grain size or orientation? Does the position of the grain in the thickness of the layer have an influence?

Reply: In the reply to Comment 15 and 16, we present the reason why it is reasonable to study the total intensity of a spot, which are actually the sum of the intensities of different indices and coming from different domains. As shown in Fig. S1 in SM, we have three in-plane 120° orientations (called 120° domains for convenience). For each domain, there are many grains with different lateral sizes. For most grains, the long side is about 50 ~ 150 nm. In nanometer scale, the film is not uniform. However, our UED beam size is about 150 μm in diameter. In such a big region, the total volume of three domains as well as the distribution of the grains in the domain should be equal. In this sense, we are studying a “uniform” film. The information we obtained is the overall structure dynamics in our detection region. For example, we observed that the superstructure peak disappeared within about 200 fs after photoexcitation. This time scale reflects the overall time scale when V-V dimers within the detect region are totally destroyed. In fact, our samples are much better than polycrystalline samples considering the domains and grains.

The growth mode of VO₂ on STO(111) is epitaxial growth, although there are three orientations due to the lattice matching between VO₂ and STO(111). The in-plane orientation of each grain will not change during the film growth. In other words, each grain is uniform in the thickness direction. Therefore, the position of the grain in the thickness of the layer is not a problem.

In the revised version, we added more discussion to clarify those issue in Fig. S1 in SM, as well as in page 7, 2nd paragraph in the main text.

Comment #19:

214. “diffraction spots” (instead of sports)

Reply: We corrected in the revised version.

Comment #20:

287. “found this new process in quasi-single (instead of signal) crystalline samples”

Reply: We corrected in the revised version.

Comment #21:

315. “roughness (instead of Thickness) of VO₂ film were checked by AFM. “How was VO₂ film thickness measured?”

Reply: We measured both the thickness and roughness of VO₂ films by AFM. We prepared more than ten samples using the same experimental parameters. The thickness and roughness are very similar. For UED experiments, we transfer the freestanding VO₂ onto the Cu grid. To check the thickness, we transferred the freestanding VO₂ to a Si substrate. The results are presented in Figs. S1 (d) and (e) in SM. The thickness of our films is about 40 nm and the roughness is about 6 nm.

Supplementary materials (SM)

Comment #22:

Crystal structure and morphology of VO₂ films.

“The epitaxial relationship is...” An X-ray diffraction experiment, for example, would highlight the epitaxy relationship between VO₂ and STO. “STO (111) has 6-fold symmetry “3-fold axis

Reply: The epitaxial relation of VO₂ on STO (111) has been studied in detail in Ref. 46 using XRD. There are three 120° domains (orientations). The epitaxy relationship between VO₂ and STO in one of the three domains is $[001](010)_{VO_2} // [01\bar{1}](111)_{STO}$ and $[100](010)_{VO_2} // [\bar{2}11](111)_{STO}$. We sketched these three domains in Fig. S1 (b) in SM. Based on the three domains, RHEED patterns can be nicely reproduced. The simulated RHEED pattern is overlaid on the experimental pattern in the revised Fig. S1 in SM. Furthermore, in our TEM measurements in Fig. 1(c) of main text, we can observe the diffraction spots from STO(111) buffer layer, which again confirms the epitaxy relation between VO₂ and STO.

Comment #23:

Simulation on the TEM diffraction pattern of VO₂ film

Figure S2c : Indexation de la phase R dans le réseau monoclinique. Please indicate the zone axis [010] in the figure. The actual indexing corresponds to the monoclinic form. To make the simulation easier to read, please indicate M (and not R) as the hkl index. Indexing some reflections in the 2 systems would help (example 002M and 200R, 40-2M and 002R).

Reply: We added [010] and corrected the index in the revised version. For easy reading, we only add indices in the M₁ phase in Figs. S1(d) and (f), since all Miller indices in the manuscript use the M₁-phase’s representation.

Comment #24:

Figure S2d : Monoclinic, so hklM.

“M1-phase has superstructure spots (small spots in Fig. S2(d)) from the V-V dimers.
“ The sentence is confusing.

Reply: We corrected the index. In the revised version, we changed this sentence to “For

convenience, we use smaller dots to present the superstructure spots.” in Fig. S2d.

Comment #25:

Lattice dynamics after Fig S4 b. The -102M reflection is almost superimposed with an STO reflection. How was the STO contribution withdrawn?

Reply: The referee is right, -102 reflection does overlap with a STO spot. For Fig. S4(b), we didn't withdraw the contribution from STO when we analysis the change of intensity of -102. Similar to 102 spot, pure -102 spot (without contribution of STO) should disappear after VO₂ changes from the M₁ to R phase. Because of overlapping with STO spot, the relative drop of intensity of -102 spot is less than 30% of that of 102 spot. Because Fig. S4(b) did not provide any new information compared with Fig. S4(a), we removed this panel in the revised Fig. S4 in SM.

Comment #26:

The relation between the peak width of 200R peaks and the twin domain. Fig S5 c, d, e
The peak width difference (c and e) is 1 pixel. Is it significant? What is the measurement uncertainty?

Reply: Thanks for the referee's careful review. The measurement uncertainty depends on the diffraction signal, stability of incident electrons, exposure time and number of measurements. In our case, we captured a diffraction pattern with an exposure time of two seconds at each time delay and repeated 45 times. Then we averaged the diffraction patterns at each time delay to get the final diffraction pattern. The width measured at a fixed time delay averaged over forty-five measurements yields an uncertainty of about 0.13 pixel. The change of width we measured is nearly ten times larger than the uncertainty, which demonstrates that the change of this signal is credible. In revised Fig. S5, we also evaluated the peak narrowing during the phase transition from the M₁ phase to R phase. The results of simulation are comparable with our experimental results.

Reviewer #3:

This paper reports an important investigation of atomic dynamics during the photo-induced phase transition (PIPT) in quasi-single-crystal VO₂ using ultrafast electron diffraction (UED) with 50 femtosecond temporal resolution and a 3 MeV electron source. Its major conclusion is that the PIPT first drives a transient monoclinic phase in which the vanadium ion chains are neither dimerized nor in a zig-zag chain configuration before settling into the quasi-equilibrium R phase about 5 ps after the initial photoexcitation event. The paper is publishable after one major and a few minor weak points are corrected.

We want to thank the referee for the very careful review and helping us improve the clarity of the manuscript. In the revised version, we corrected all weak points.

Comment #1:

Weaknesses, corrections and suggestions: The major weakness in the paper is the repeated assertion that the new measurements do not support the existence of the monoclinic metallic (mM) state reported in References 18, 32 and 35. Because the present work presents only diffraction data, the only basis for this assertion seems to be the duration of the monoclinic transient and its lack of a second excitation threshold. However, diffraction measurements alone do not yield measurements of free-electron density that confirm metallicity. (See, for example, the side-by-side measurements of diffracted intensity and THz conductivity presented in Fig 4 of Ref. 35.). Thus references to the mM phase (for which there is solid evidence in LEEM experiments) should be omitted.

Reply: We agree with the referee that diffraction measurements alone do not yield measurements of free-electron density that confirm metallicity. I totally agree that there should be some mM phase in VO₂. LEEM experiment suggests a mM phase in thermal equilibrium. Very recent calculations suggest a mM phase can exist in photoinduced transition in the first 10- 40 fs [Sci. Adv. 8, eadd2392 (2022)]. Ref. 18 suggested another kind of photoinduced mM phase with a very long lifetime (sever hundreds of picoseconds). One of the key evidences in ref. 18 is two thresholds for the superstructure

peak and non-superstructure peak. In our films, we don't have resolution to check whether mM phase exists in the first 10 – 40 fs. Since we have excellent signal-to-noise ratio in the diffraction pattern, we can determine the threshold accurately. Therefore, we carried our threshold measurement. Our results show that only one threshold was observed in our films. We can only say that in our film we don't have such type of mM phase as proposed in ref. 18. In the revised version, we revised our statements to explicitly clarify this issue. We would like to state in the revised manuscript that “*It should be noted that the diffraction measurements do not yield the direct measurements of free-electron density that confirm metallicity. Ref. 18 suggested an interesting photoinduced mM phase with a very long lifetime (sever hundreds of picoseconds). Besides the optical measurements, one of the key evidences is the existence of two thresholds in UED. Since only one threshold is detected in our experiments, the proposed long lifetime mM phase in Ref. 18 very likely does not exist in our quasi-single crystalline samples.*”. In the future, we plan to carry out optical spectroscopic to check possible photoinduced metallic phase in our films.

Comment #2:

Because the paper makes comparisons both to bulk single-crystal and polycrystalline thin-film data, it would be helpful if the authors would spell out precisely what they mean by “quasi-single-crystal.” It sounds like all that is intended is that their samples are well lattice-matched with some twinning, large domain sizes and a clearly defined film plane as suggested by the TEM simulation in Fig. 1(d); if that is indeed the case they should also give the lattice parameters and matching statistics. It would also be useful to state the lateral dimensions in the main text.

Reply: The referee is right. Our samples are indeed well lattice-matched and have a clearly defined film plane. The normal direction to the film surface is [010]. In the plane, VO₂ has three in-plane orientations with an interval of 120° because of the lattice matching between VO₂ and the STO (111) buffer layer. In the main text of revised manuscript, we gave the clear meaning of “quasi-single crystalline”. Lattice parameters,

matching statistics and epitaxial relation was added in the description of Fig. S1 in SM. The Lateral dimensions was added in the main text, as well as in Fig. S1 in SM.

Comment #3:

In commenting on Ref. 37 (page 3) the authors wrongly characterize the VO₂ sample as having been “fabricated by focused ion beam” (FIB), when in fact the sample is a bulk single crystal that has been thinned by FIB.

Reply: We corrected it in the revised version.

Comment #4:

Style and mechanics issues: The title is so long that it almost seems like an abbreviated abstract, and yet does not include the essential information that the transient monoclinic phase is generated by photoexcitation! I suggest something like “Transient dynamics of the phase transition in VO₂ by ultrafast 3 MeV electron diffraction.”

Reply: Thanks for the referee’s very helpful suggestion. Following the referee’s suggestion, we have changed the title to “Transient dynamics of the phase transition in quasi-single-crystal VO₂ by MeV ultrafast electron diffraction”.

Comment #5:

The paragraphs on pages 8 and 9 of the manuscript are so long that the thread of analysis is difficult to follow. The authors need to break up these paragraphs so that each new point in the discussion is allotted a separate paragraph.

Reply: We corrected this issue in the revised version.

Comment #6:

In Figure 1(a), it is difficult to see the black type against the dark colors of the layers in the sample; indeed, the SAO is completely invisible. The type should be yellow or white.

Reply: We changed the color to white.

Comment #7:

Apart from occasional lapses in wording that indicate that the authors are not native English speakers, there are many misspellings, some of which actually muddle the intended meaning. It would be advisable to have the manuscript proofread by a native English speaker.

Reply: Thanks for referee's suggestion. We had the revised version proofread by a native English speaker to help us improve the English.

REVIEWERS' COMMENTS

Reviewer #1 (Remarks to the Author):

The improvements in revised manuscripts made by authors includes quantitative analysis of data and thus seems reasonable to support their findings and conclusion, which are important for material science community. Therefore I think it is worthy of publication in Nature Communications. All I suggest in the following are descriptions (including minor typos) to be revised.

- page 3 of the main text: "considerable large" -> "considerably large"
- page 11 of the main text: "the diffraction measurements do not yield the direct measurements of free-electron density" -> "the diffraction measurements do not yield the direct information of free-electron density"
- page 7 of the main text: "The structure fact of 102 and $3\bar{0}2$ are zero in the R phase" -> "The structure factor of 102 and $3\bar{0}2$ are zero in the R phase"

Reviewer #2 (Remarks to the Author):

The authors have greatly improved the overall quality of the main document as well as that of the supplementary material by adding relevant elements (figures, arguments, discussions). They also answered in a detailed and convincing way to all the remarks or questions. I support the publication of this manuscript.

Below are our point-to-point responses to reviewers' comments on our manuscript.

Reviewer #1

The improvements in revised manuscripts made by authors includes quantitative analysis of data and thus seems reasonable to support their findings and conclusion, which are important for material science community. Therefore, I think it is worthy of publication in Nature Communications. All I suggest in the following are descriptions (including minor typos) to be revised.

- page 3 of the main text: "considerable large" -> "considerably large"
- page 11 of the main text: "the diffraction measurements do not yield the direct measurements of free-electron density" -> "the diffraction measurements do not yield the direct information of free-electron density"
- page 7 of the main text: "The structure fact of 102 and $3\bar{0}2$ are zero in the R phase" -> "The structure factor of 102 and $3\bar{0}2$ are zero in the R phase"

Reply: We thank the referee for his/her careful review and helping us to improve our manuscripts. We corrected these typos in the revised manuscripts.

Reviewer #2

The authors have greatly improved the overall quality of the main document as well as that of the supplementary material by adding relevant elements (figures, arguments, discussions). They also answered in a detailed and convincing way to all the remarks or questions. I support the publication of this manuscript.

Reply: We appreciate that the referee supports the publication of this manuscript in Nature Communications.